# Optimizing protein fitness using Bi-level Gibbs sampling with Graph-based Smoothing

## Abstract

The ability to design novel proteins with higher fitness on a given task would be revolutionary for many fields of medicine. However, brute-force search through the combinatorially large space of sequences is infeasible. Prior methods constrain search to a small mutational radius from a reference sequence, but such heuristics drastically limit the design space. Our work seeks to remove the restriction on mutational distance while enabling efficient exploration. We propose **Bi**-level **G**ibbs sampling with **G**raph-based **S**moothing (BiGGS), which uses the gradients of a trained fitness predictor to sample many mutations towards higher fitness. Bi-level Gibbs first samples sequence locations then sequence edits. We introduce graph-based smoothing to remove noisy gradients that lead to false positives. Our method is state-of-the-art in discovering high-fitness proteins with up to 8 mutations from the training set. We study the GFP and AAV design problems, ablations, and baselines to elucidate the results.

## 1 Introduction

In protein design, fitness is loosely defined as performance on a desired property or function. Examples of fitness include catalytic activity for enzymes [1, 20] and fluorescence for biomarkers [27]. Protein engineering seeks to design proteins with high fitness by altering the underlying sequences of amino acids. However, the number of possible proteins increases exponentially with sequence length, rendering it infeasible to perform brute-force search to engineer novel functions which often requires many mutations (i.e. at least 3 [11]). Directed evolution [3] has been successful in improving protein fitness, but it requires substantial labor and time to gradually explore many mutations.

We aim to find shortcuts to generate high-fitness proteins that are many mutations away from what is known but face several challenges. Proteins are notorious for highly non-smooth fitness landscapes:[1] fitness can change dramatically with just a single mutation, and most protein sequences have zero fitness [29]. As a result, machine learning (ML) methods are susceptible to learning noisy fitness landscapes with false positives [18] and local optimums [6] which poses problems to optimization and search. The 3D protein structure, if available, can help provide helpful constraints in navigating the noisy fitness landscape, but it cannot be assumed in the majority of cases – current protein folding methods typically cannot predict the effects on structure of point mutations [25].

Our work proposes a sequence-based method that can optimize over a noisy fitness landscape and efficiently sample large mutational edits. We introduce two methodological advances summarized in Figure 1. The first is Graph-based Smoothing (GS) that regularizes the noisy landscape. We consider it as a noisy graph signal and apply $L_1$ graph Laplacian regularization. This encourages sparsity and local consistency in the landscape; most protein sequences have zero fitness, and similar sequences

---

[1]Landscape refers to the mapping from sequence to fitness.

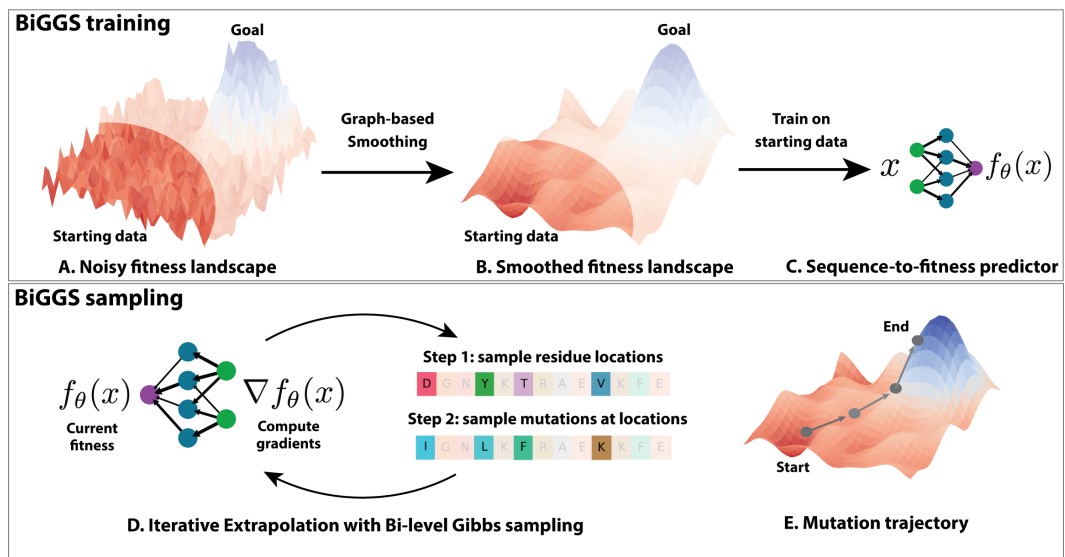

Figure 1: BiGGS overview. **(A)** Protein engineering is often challenged with a noisy fitness landscape on which the starting dataset (unblurred) is a fraction of landscape with the highest fitness sequences hidden (blurred). **(B)** We develop Graph-based Smoothing (GS) to estimate a smoothed fitness landscape from the starting data. Intuitively, the gradients allow extrapolation towards higher fitness sequences. **(C)** A fitness predictor is trained on the smoothed fitness landscape. **(D)** Gradients from the fitness predictor are used in an iterative sampling procedure called Iterative Extrapolation (IE) where Bi-level Gibbs sampling (BiG) is performed on each step with renewed gradient computations. **(E)** Each round of IE samples mutations towards higher fitness.

have similar fitness [42]. The effect is a smooth fitness landscape learned by the ML model on which gradients accurately approximate the direction towards high-fitness sequences. To reach high-fitness sequences requiring many mutations, we use the improved gradients in our second advancement, Bi-level Gibbs (BiG), to approximate the proposal distribution in a Gibbs sampling procedure – as inspired by Gibbs with Gradients (GWG) [12]. BiG uses bi-level sampling to propose up to 5 indices to mutate simultaneously. Local improvements from the gradients help select beneficial mutations to guide low-fitness sequences towards higher fitness while sampling allows exploration. Following the intuition of directed evolution, we apply multiple rounds of sampling over clustered sequences in a procedure we call Iterative Extrapolation (IE).

We find BiG and GS are complementary in enabling long-range exploration while avoiding the pitfalls of a noisy fitness landscape; the combination of both is referred to as BiGGS. We introduce a set of tasks using the Green Fluorescent Proteins (GFP) dataset [30] to simulate challenging protein design scenarios by starting with low-fitness sequences that require many (5 or more) mutations to the best fitness. We primarily study GFP because of (1) its difficulty as one of the longest proteins in fitness datasets and (2) its comprehensive fitness measurements of up to 15 mutations. To assess the generalizability of our method, we additionally study the Adeno-Associated Virus (AAV) dataset [7] based on gene delivery fitness. We evaluate BiGGS and prior works on our proposed benchmarks to show that BiGGS is state-of-the-art in GFP and AAV fitness optimization. Our contributions are summarized as follows:

- We develop a novel sequence-based protein fitness optimization algorithm, BiGGS, based on BiG to efficiently sample multiple mutations, GS to regularize the fitness landscape, and IE to progressively mutate towards higher-fitness (Section 2).

- We study GFP by proposing a set of design benchmarks of different difficulty with varying starting sequence distribution (Section 3). While our focus is GFP, we develop benchmarks on AAV to evaluate a new fitness criteria (Appendix C).

- We show BiGGS is state-of-the-art in GFP and AAV fitness optimization while exhibiting diversity and novelty from the training set. We analyze the contributions of BiG and GS towards successful fitness optimization over challenging fitness landscapes (Section 5).

 ## 2 Method

 We begin with the problem formulation in Section 2.1. Our method uses two components bi-level
 Gibbs sampling (Section 2.2) and graph-based smoothing (Section 2.3). Together they are part of
 a iterative sampling method called iterative extrapolation (Section 2.4) as a way to progressively
 extrapolate towards novel sequences. The full algorithm, BiGGS, is presented in Algorithm 1.

### 2.1 Problem formulation

Let the starting set of length $L$ protein sequences and their fitness measurements be denoted as
$\mathcal{D}_0 = (\mathcal{X}_0, \mathcal{Y}_0)$ where $\mathcal{X}_0 \subset \mathcal{V}^L$ with vocabulary $\mathcal{V} = \{1, \ldots, 20\}$ and $\mathcal{Y}_0 \subset \mathbb{R}$. We use subscripts
to distinguish sequences, $x_i \in \mathcal{V}^L$, while a paranthetical subcript denotes the token, $(x_i)_j \in \mathcal{V}$ where
$j \in \{1, \ldots, L\}$. Note our method can readily be extended to other modalities, e.g. nucleic acids.

For *in-silico* evaluation, we denote the set of *all* known sequences and fitness measurements as
$\mathcal{D}^* = (\mathcal{X}^*, \mathcal{Y}^*)$. We assume there exists a black-box function $g : \mathcal{V}^L \to \mathbb{R}$ such that $g(x^*) = y^*$,
which is approximated by an oracle $g_\phi$. In practice, the oracle is a model trained with weights $\phi$ to
minimize prediction error on $\mathcal{D}^*$. The starting dataset only includes low fitness sequences and is a
strict subset of the oracle dataset $\mathcal{D}_0 \subset \mathcal{D}^*$ to simulate fitness optimization scenarios. Given $\mathcal{D}_0$, our
task is to generate a set of sequences $\hat{\mathcal{X}} = \{\hat{x}_i\}_{i=1}^{N_{\text{samples}}}$ with higher fitness than the starting set.

### 2.2 BiG: Bi-level Gibbs (with Gradients)

To generate new sequences, we propose a modified version of Gibbs With Gradients (GWG) [12].
The first step is to train a fitness *predictor*, $f_\theta : \mathcal{V}^L \to \mathbb{R}$, using $\mathcal{D}_0$ to act as the learned unnormalized
probability (i.e. negative energy) from sequence to fitness. We use the Mean-Squared Error (MSE)
loss to train the predictor which we parameterize as a deep neural network. We found it beneficial to
employ negative data augmentation since both the dataset and the range of fitness values are small.
Specifically, we double the size of the dataset by sampling random sequences, $x_i^{\text{neg}} \sim \text{Uniform}(\mathcal{V}^L)$,
and assigning them the lowest possible fitness value, $\mu$.

Our goal is to sample from $\log p(x) = f_\theta(x) - \log Z$ where $Z$ is the normalization constant. Higher
fitness sequences will be more likely under this distribution while sampling over many mutations will
induce diversity and novelty. GWG uses Gibbs sampling with *locally informed proposals*:

$$q^\nabla(x'|x) \propto e^{\frac{(x')^\top d_\theta(x)}{2}} \mathbb{1}(x' \in H(x)), \qquad d_\theta(x)_{ij} = \nabla_x f_\theta(x)_{ij} - x_i^T \nabla f_\theta(x)_i, \qquad (1)$$

where $d_\theta(x)_{ij}$ is a first order Taylor approximation of the log-likelihood ratio of mutating the $i$th
index of $x$ to token $j$. Treating $x, x'$ as one-hot, $(x')^\top d_\theta(x) = \sum_i (x_i')^\top d_\theta(x)_i$ is the sum over the
local differences where $x'$ differs from $x$. The proposal $q(x'|x)$ can be efficiently computed when
$H(\cdot)$ is the 1-Hamming ball[2]: a single backward pass is needed to compute the Jacobian in eq. (1).

Sampling $M > 1$ mutations in the same fashion would require estimating the gradients for each
mutation individually resulting in exponentially more computations. Instead, we find a simple bi-level
sampling scheme to be effective. The first level samples mutation indices, $\ell_m$, with a categorical
tempered-softmax distribution over the column-wise maxima, $d_\theta(x)_i = \max_{j \in \{1,\ldots,L\}} d_\theta(x)_{ij}$. The
second level samples token-wise mutations $(x')_{\ell_m}$ over the vocabulary the same way as the first level
using $d_\theta(x)_{\ell_m j}$.

$$\textbf{First level}: \ell_m \overset{iid}{\sim} q(\cdot|x) = \text{Cat}\left(\text{Softmax}\left(\left\{\frac{d_\theta(x)_i}{\tau}\right\}_{i=1}^L\right)\right), \quad m \in \{1, \ldots, M\}$$

$$\textbf{Second level}: (x')_{\ell_m} \sim q(\cdot|x, \ell_m) = \text{Cat}\left(\text{Softmax}\left(\left\{\frac{d_\theta(x)_{\ell_m j}}{\tau}\right\}_{j=1}^{|\mathcal{V}|}\right)\right)$$

$$(2)$$

where $\tau$ is a temperature hyperparameter. Indices are sampled $iid$ which means the same index may
get sampled twice. An improvement left for future work is to model conditional dependencies across

---

[2]Defined as a ball using the hamming distance.

locations. Each proposed sequence is accepted or rejected using Metropolis-Hasting (MH)

$$\min\left(\exp(f_\theta(x') - f_\theta(x))\frac{\prod_{m=1}^M q((x')_{\ell_m}|x, \ell_m)q(\ell_m|x)}{\prod_{m=1}^M q((x)_{\ell_m}|x', \ell_m)q(\ell_m|x')},\ 1\right). \tag{3}$$

To summarize, our method Bi-level Gibbs (BiG) first samples $N_{\text{prop}}$ sequences each with up to $M$ mutations from eq. (2) then returns a set of accepted sequences, $\mathcal{X}'$, according to eq. (3). Forcing BiG to make $M$ mutations may make it skip sequences that are less than $M$ mutations away. We found it best to run BiG over all values leading up to $M$. The full algorithm is provided in algorithm 2.

A concern is the accuracy of the 1st order Taylor approximation, $d_\theta(x)_{ij}$, for $M > 1$. We observed the performance of BiG is highly dependent on the performance of the predictor for gradients that correlate with higher fitness. The next two sections focus on the development of a robust predictor (Section 2.3) and an iterative framework to improve the Gibbs sampling approximation (Section 2.4).

## 2.3   GS: Graph-based smoothing

The efficacy of the gradients in BiG to guide sampling towards high fitness sequences depends on the smoothness of the mapping from sequence to fitness learned by the predictor. Unfortunately, the high-dimensional sequence space coupled with few data points and noisy labels results in a noisy predictor that is prone to sampling false positives [18] or getting stuck in local optima [6]. To address this, we use techniques from graph signal processing to smooth the learned mapping by promoting similar sequences to have similar fitness [42] while penalizing noisy predictions [17].

Suppose we have trained a noisy predictor with weights $\theta_0$ on the initial dataset $\mathcal{D}_0$. To construct our graph $G = (V, E)$, we first construct the nodes $V$ by iteratively applying pointwise mutations to each sequence in the initial set $\mathcal{X}_0$ to simulate a local landscape around each sequence. We call this routine `Perturb` with a hyperparameter $N_{\text{perturb}}$ for the number of perturbations per sequence (see Algorithm 5). The edges, $E$, are a nearest neighbor graph with $N_{\text{neigh}}$ neighbors where edge weights are inversely proportional to their sequence distance, $\omega_{ij} = \omega((v_i, v_j)) = 1/\texttt{dist}(v_i, v_j)$; edge weights are stored in a similarity matrix $W = \{\omega_{ij}\ \forall v_i, v_j \in V\}$.

The normalized Laplacian matrix of $G$ is $\mathcal{L} = I - D^{-1/2}WD^{-1/2}$ where $I$ is the identity and $D$ is a diagonal matrix with $i$-th diagonal element $D_{ii} = \sum_j \omega_{ij}$. An eigendecomposition of $\mathcal{L}$ gives $\mathcal{L} = U\Sigma U^T$ where $\Sigma$ is a diagonal matrix with sorted eigenvalues along the diagonal and $U$ is a matrix of corresponding eigenvectors along the columns. An equivalent eigendecomposition with symmetric matrix $B$ (and edges $E$ arranged into an adjacency matrix) is

$$\mathcal{L} = (\Sigma^{1/2}U^T)^T\Sigma^{1/2}U^T = B^TB, \qquad B = \Sigma^{1/2}U^T.$$

Next, we formulate smoothing as an optimization problem. For each node, we predict its fitness $\mathcal{S} = \{f_{\theta_0}(v)\ \forall v \in V\}$, also called the graph *signal*, which we assume to have noisy values. Our goal is to solve the following where $\mathcal{S}$ is arranged as a vector and $\mathcal{S}^*$ is the smoothed signal,

$$\mathcal{S}^* = \arg\min_{\hat{\mathcal{S}}} \|B\hat{\mathcal{S}}\|_1 + \gamma\|\hat{\mathcal{S}} - \mathcal{S}\|_1 \tag{4}$$

Equation (4) is a form of graph Laplacian regularization that has been studied for image segmentation with weak labels [17]. $B$ has eigenvalue weighted eigenvectors as rows. Due to the $L_1$-norm $\|B\hat{\mathcal{S}}\|_1$ is small if $\hat{\mathcal{S}}$ is primarily aligned with slowly varying eigenvectors whose eigenvalues are small. This term penalizes large jumps in fitness between neighboring nodes hence we call it *smoothness sparsity constraint*. The second term, $\|\hat{\mathcal{S}} - \mathcal{S}\|_1$, is the *signal sparsity constraint* that remove noisy predictions with hyperparameter $\gamma$. The $L_1$-norm is applied to reflect that most sequences have zero fitness.

At a high level, eq. (4) is solved by introducing auxiliary variables which allows for an approximate solution by solving multiple LASSO regularization problems [34]. Technical details and algorithm are described in Appendix B. Once we have $\mathcal{S}^*$, we retrain our predictor with the smoothed dataset $\mathcal{D} = (V, \mathcal{S}^*)$ on which the learned predictor is smoother with gradients much more amenable for gradient-based sampling, BiG. We refer to our smoothing algorithm as Graph-based Smoothing (GS).

## 2.4   IE: Iterative Extrapolation

The 1st order Taylor approximation of eq. (1) deteriorates the more we mutate from the parent sequence. Inspired by directed evolution [3], we propose to alleviate this by performing multiple

rounds of sampling where successive rounds use sequences from the previous round. Each round re-centers the Taylor approximation and extrapolates from the previous round. We first train a predictor $f_\theta$ using GS (Section 2.3). Prior to sampling, we observe the number of sequences may be large and redundant. To reduce the number of sequences, we perform hierarchical clustering [22] and take the sequence of each cluster with the highest fitness using $f_\theta$. Let $\mathcal{C}$ be the number of clusters.

$$\texttt{Reduce}(\{\mathcal{X}^c\}_{c=1}^{\mathcal{C}};\theta) = \bigcup_{c=1}^{\mathcal{C}}\{\arg\max_{x\in\mathcal{X}^c} f_\theta(x)\} \text{ where } \{\mathcal{X}^c\}_{c=1}^{\mathcal{C}} = \texttt{Cluster}(\mathcal{X};\mathcal{C}).$$

Each round $r$ reduces the sequences from the previous round and performs BiG sampling.

$$\mathcal{X}'_{r+1} = \bigcup_{x\in\tilde{\mathcal{X}}_r} \texttt{BiG}(x;\theta), \quad \tilde{\mathcal{X}}_r = \texttt{Reduce}(\{\mathcal{X}^c_r\}_{c=1}^{\mathcal{C}};\theta), \quad \{\mathcal{X}^c_r\}_{c=1}^{\mathcal{C}} = \texttt{Cluster}\mathcal{X}'_{r+1}(\mathcal{X}'_r;\mathcal{C}).$$

One cycle of clustering, reducing, and sampling is a round of extrapolation,

$$\mathcal{X}'_{r+1} = \texttt{Extrapolate}(\mathcal{X}'_r;\theta,\mathcal{C}) \tag{5}$$

where the initial round $r = 0$ starts with $\mathcal{X}'_0 = \mathcal{X}_0$. After $R$ rounds, we select our candidate sequences by taking the Top-$N_{\text{samples}}$ sequences based on ranking with $f_\theta$. We call this procedure Iterative Extrapolation (IE). While IE is related to previous directed evolution methods [31], it differs by taking larger mutational edits on each round with BiG and encouraging diversity by mutating the best sequence of each cluster. The full candidate generation, Bi-level Gibbs with Graph-based Smoothing (BiGGS), with IE is presented in Algorithm 1.

---

**Algorithm 1** $\texttt{BiGGS}$: Bi-level Gibbs with Graph-based Smoothing

---

**Require:** Starting dataset: $\mathcal{D}_0 = (\mathcal{X}_0, \mathcal{Y}_0)$
**Require:** BiG hyperparameters: $N_{\text{prop}}, \tau, M$
**Require:** GS hyperparameters: $N_{\text{neigh}}, N_{\text{perturb}}, \gamma$
**Require:** IE hyperparameters: $N_{\text{samples}}, R, \mathcal{C}$
1: $\mathcal{D} \leftarrow \mathcal{D}_0 \cup \{(x_i^{\text{neg}}, \mu)\}_{i=1}^{|\mathcal{D}_0|}$                        ▷ Construct negative data
2: $\theta_0 \leftarrow \arg\max_{\tilde{\theta}} \mathbb{E}_{(x,y)\sim\mathcal{D}} \left[(y - f_{\tilde{\theta}}(x))^2\right]$            ▷ Initial training.
3: $\theta \leftarrow \texttt{Smooth}(\mathcal{X}_0;\theta_0)$                               ▷ GS Algorithm 3.
4: $\{\mathcal{X}_0\}_{c=1}^{\mathcal{C}} \leftarrow \texttt{Cluster}(\mathcal{X}_0;\mathcal{C})$                   ▷ Initial round of IE
5: $\tilde{\mathcal{X}}_0^c \leftarrow \texttt{Reduce}(\{\mathcal{X}_0\}_{c=1}^{\mathcal{C}};\theta)$
6: $\mathcal{X}'_0 \leftarrow \cup_{x\in\tilde{\mathcal{X}}_0^c}\texttt{BiG}(x;\theta)$                      ▷ BiG algorithm 2
7: **for** $r = 1,\ldots,R$ **do**
8:     $\mathcal{X}'_r \leftarrow \texttt{Extrapolate}(\mathcal{X}'_{r-1};\theta)$          ▷ Remaining rounds of IE eq. (5)
9: **end for**
10: $\hat{\mathcal{X}} \leftarrow \texttt{TopK}(\cup_{r=1}^R \mathcal{X}'_r)$       ▷ Return Top-$N_{\text{samples}}$ sequences based on predicted fitness $f_\theta$.
11: **Return** $\hat{\mathcal{X}}$

---

# 3 Benchmarks

We use the Green Fluoresent Protein (GFP) dataset from Sarkisyan et al. [30] containing over 56,806 log fluorescent fitness measurements, with 51,715 unique amino-acid sequences due to *sequences having multiple measurements*. We quantify the difficulty of a protein fitness optimization task by introducing the concept of a *mutational gap*, which we define as the minimum Levenshtein distance between any sequence in the training set to any sequence in the 99th percentile:

$$\text{Gap}(\mathcal{X}_0; \mathcal{X}^{\text{99th}}) = \texttt{min}(\{\texttt{dist}(x, \tilde{x}) : x \in \mathcal{X}, \tilde{x} \in \mathcal{X}^{\text{99th}}\})$$

A mutational gap of 0 means that the training set, $\mathcal{D}_0$ may contain sequences that are in the 99th percentile of fitness. Solving such tasks is easy because methods may sample high-fitness sequences from the training set. Prior work commonly uses the GFP task introduced by design-bench (DB) evaluation framework [36] which has a mutational gap of 0 (see Appendix A). To compare to previous work, we include the DB task as **"easy"** difficulty in our experiments, but we introduce **"medium"** and **"hard"** optimization tasks which have lower starting fitness ranges in the 20-40th and 10-30th percentile of known fitness measurements alongside much higher mutational gaps. Our proposed difficulties are summarized in Table 1 and visualized in Figure 5.

The oracle in design-bench (DB) uses a Transformer-based architecture from Rao et al. [26]. When using this oracle, we noticed a concerning degree of false positives and a thresholding effect of its predictions. We propose a simpler CNN architecture as the oracle that achieves superior performance in terms of Spearman correlation and fewer false positives as seen in Figure 6. Our CNN

Table 1: Proposed GFP tasks

| Difficulty | Range (%) | $|\mathcal{D}_0|$ | Gap |
|---|---|---|---|
| Medium | 20th-40th | 2828 | 6 |
| Hard | 10th-30th | 1636 | 7 |

consists of a 1D convolutional layer that takes in a one-hot encoded sequence, followed by max-pooling and a dense layer to a single node that outputs a scalar value. It uses 256 channels throughout for a total of 157,000 parameters – 15 fold fewer than DB oracle.

Our experiments in Section 5 benchmark on GFP easy, medium, and hard with our CNN oracle. In Appendix C we summarize an additional benchmark using Adeno-Associated Virus (AAV) dataset [7] which focuses on optimizing a 28-amino acid segment for DNA delivery. We use the same task set-up and train our CNN oracle on AAV.

# 4   Related work

**Optimization in protein design.** Approaches in protein design can broadly be categorized in using sequence, structure or both [9]. Advances in structure-based protein design have been driven by a combination of geometric deep learning and generative models [37, 13, 39, 8]. Sequence-based protein design has been explored through the lens of reinforcement learning [2, 16], latent space optimization [32, 16, 19], GFlowNets [14], bayesian optimization [38], generative models [6, 5, 23, 21], and model-based directed evolution [31, 4, 24, 28, 35]. Together they face the common issue of a noisy landscape to optimize over. Moreover, fitness labels are problem-dependent and scarce, apart from well-studied proteins [5]. Our method addresses small amounts of starting data and noisy landscape by regularization with GS. We focus on sequence-based methods where we use locally informed Markov Chain Monte Carlo (MCMC) methods [40] method based on Gibbs With Gradients (GWG) [12] which requires a smooth energy function for strong performance guarantees. Concurrently, Emami et al. [10] used GWG to sample higher fitness sequences by optimizing over a product of experts distribution, a mixture of a protein language model and a fitness predictor. However, they eschewed the need for a smooth energy function which we address with GS.

**Discrete MCMC.** High-dimensional discrete MCMC can be inefficient with slow mixing times. GWG showed discrete MCMC becomes practical by utilizing learned gradients in the sampling distribution, but GWG in its published form was limited to sampling in a proposal window of size 1. Zhang et al. [41] proposed to modify GWG with langevin dynamics to allow for the whole sequence to mutate on every step while Sun et al. [33] augmented GWG with a path auxiliary proposal distribution to propose a series of local moves before accepting or rejecting. We find that BiGGS with bi-level sampling is simpler and effective in achieving a proposal window size beyond 1.

# 5   Experiments

We study the performance of BiGGS on the GFP tasks from Section 3. Furthermore, to ensure that we did not over-optimize to the GFP dataset, we benchmark BiGGS using AAV benchmark in Appendix C. In the subsequent sections, we outline our experiments on GFP, while corresponding results for AAV are in Appendix C. Section 5.1 compares the performance of BiGGS on GFP to a representative set of baselines while Section 5.2 performs ablations on components of BiGGS. Finally, Section 5.3 analyzes BiGGS's performance.

**BiGGS training and sampling.** Following section 3, we use the oracle CNN architecture for our predictor (but trained on different data). To ensure a fair comparison, we use the same predictor across all model-based baselines. We use the following hyperparameters as input to Algorithm 1 across all tasks: $N_{\text{prop}} = 100$, $\tau = 0.01$, $M = 5$, $N_{\text{neigh}} = 500$, $N_{\text{perturb}} = 1000$ $N_{\text{samples}} = 128$ $R = 3, \mathcal{C} = 500$. We were unable to perform extensive exploration of hyperparameters. Reducing the number of hyperparameters and finding optimal values is an important future direction. Training is performed with batch size 1024, ADAM optimizer [15] (with $\beta_1 = 0.9, \beta_2 = 0.999$), learning rate 0.0001, and 1000 epochs using a single A6000 Nvidia GPU. Initial predictor training takes 10 minutes while graph-based smoothing takes around 30 minutes depending on convergence of the

numerical solvers. Training with the smoothed data takes 4 to 8 hours. Sampling takes under 30 minutes and can be parallelized.

**Baselines.** We choose a representative set of prior works with publicly available code: GFlowNets (GFN-AL) [14], model-based adaptive sampling (CbAS) [6], greedy search (AdaLead) [31], bayesian optimization with quasi-expected improvement acquisition function (BO-qei) [38], conservative model-based optimization (CoMs) [35], and proximal exploration (PEX) [28].

**Metrics.** Each method generates $N_{\text{samples}} = 128$ samples $\hat{\mathcal{X}} = \{\hat{x}_i\}_{i=1}^{N_{\text{samples}}}$ to evaluate. Here, `dist` is the Levenshtein distance. We report three metrics:

- **(Normalized) Fitness** = $\texttt{median}(\{\xi(\hat{x}_i; \mathcal{Y}^*)\}_{i=1}^{N_{\text{samples}}})$ where $\xi(\hat{x}; \mathcal{Y}^*) = \frac{g_\phi(\hat{x}_i) - \min(\mathcal{Y}^*)}{\max(\mathcal{Y}^*) - \min(\mathcal{Y}^*)}$ is the min-max normalized fitness.

- **Diversity** = $\texttt{mean}(\{\texttt{dist}(x, \tilde{x}) : x, \tilde{x} \in \hat{\mathcal{X}}, x \neq \tilde{x}\})$ is the average sample similarity.

- **Novelty** = $\texttt{median}(\{\eta(\hat{x}_i; \mathcal{X}_0)\}_{i=1}^{N_{\text{samples}}})$ where $\eta(x; \mathcal{X}_0) = \min(\{\texttt{dist}(x, \tilde{x}) : \tilde{x} \in \mathcal{X}^*, \tilde{x} \neq x\})$ is the minimum distance of sample $x$ to any of the starting sequences $\mathcal{X}_0$.

We use median for outlier robustness. Diversity and novelty were introduced in Jain et al. [14]. We emphasize that higher diversity and novelty is *not* equivalent to better performance. For instance, a random algorithm would achieve maximum diversity and novelty.

## 5.1 Results

All methods are evaluated on 128 generated candidates, as done in design-bench. We run 5 seeds and report the average metric across all seeds including the standard deviation in parentheses. Results using our GFP oracle are summarized in table 2. Results using the DB oracle are in appendix C.

Table 2: GFP optimization results (our oracle).

| GFP Task | | Method | | | | | | |
|---|---|---|---|---|---|---|---|---|
| Difficulty | Metric | GFN-AL | CbAS | Adalead | BO-qei | CoMs | PEX | **BiGGS** |
| Easy | Fit. | 0.16 (0.0) | 0.81 (0.0) | **0.92 (0.0)** | 0.77 (0.0) | 0.06 (0.3) | 0.71 (0.0) | **0.92 (0.0)** |
| | Div. | 27.9 (2.0) | 4.5 (0.4) | 2.1 (0.2) | 5.9 (0.0) | 129 (16) | 2.2 (0.1) | 2.2 (0.0) |
| | Nov. | 215 (2.9) | 1.4 (0.5) | 1.0 (0.0) | 0.0 (0.0) | 164 (80) | 1.0 (0.0) | 1.0 (0.0) |
| Medium | Fit. | 0.13 (0.0) | 0.21 (0.0) | 0.53 (0.0) | 0.17 (0.0) | -0.1 (0.0) | 0.51 (0.0) | **0.86 (0.0)** |
| | Div. | 30.9 (2.7) | 9.2 (1.5) | 9.3 (0.1) | 20.1 (7.1) | 142 (15.5) | 2.0 (0.0) | 4.0 (0.2) |
| | Nov. | 214 (3.3) | 7.0 (0.7) | 1.0 (0.0) | 0.0 (0.0) | 190 (10.5) | 1.0 (0.0) | 5.9 (0.2) |
| Hard | Fit. | 0.17 (0.0) | -0.08 (0.0) | 0.03 (0.0) | 0.01 (0.0) | -0.1 (0.2) | -0.11 (0.0) | **0.43 (0.0)** |
| | Div. | 29.3 (2.2) | 98.7 (16) | 6.6 (0.6) | 84.0 (7.1) | 140 (7.1) | 2.0 (0.0) | 4.1 (0.1) |
| | Nov. | 212 (2.0) | 46.2 (9.4) | 1.0 (0.0) | 0.0 (0.0) | 198 (2.9) | 1.0 (0.0) | 7.0 (0.0) |

BiGGS substantially outperforms other baselines on the medium and hard difficulties, consistently navigating the mutational to achieve high fitness, while maintaining diversity and novelty from the training set. The unique extrapolation capabilities of BiGGS on the hardest difficulty level warranted additional analysis, and we investigate this further in Section 5.3. Adalead overall performed second-best, matching the performance of BiGGS on the easy difficulty with PEX only slightly worse. Notably, both Adalead and PEX suffer from a low novelty in the medium and hard settings.

Regarding the other baselines, GFN-AL exhibits subpar performance across all difficulty levels. We were unable to reproduce their published results.[3] Its performance notably deteriorates on medium and hard difficulty levels, a trend common amongst all baselines. CbAS explores very far, making on average 46 mutations, resulting in poor fitness. BO-qei is unable to extrapolate beyond the training set, and CoMs presents instability, as indicated by their high standard deviations, and collapse.[4]

We further analyze the distribution of novelty and fitness among CbAS, Adalead, and our method, BiGGS, in Figure 2. Adalead tends to be conservative, while CbAS is excessively liberal. BiGGS, on

---

[3]We contacted the authors but there was no resolution. Lee et al. [16] also were unable to reproduce GFN-AL.
[4]CoMs managed to generate only between 7 and 65 unique sequences.

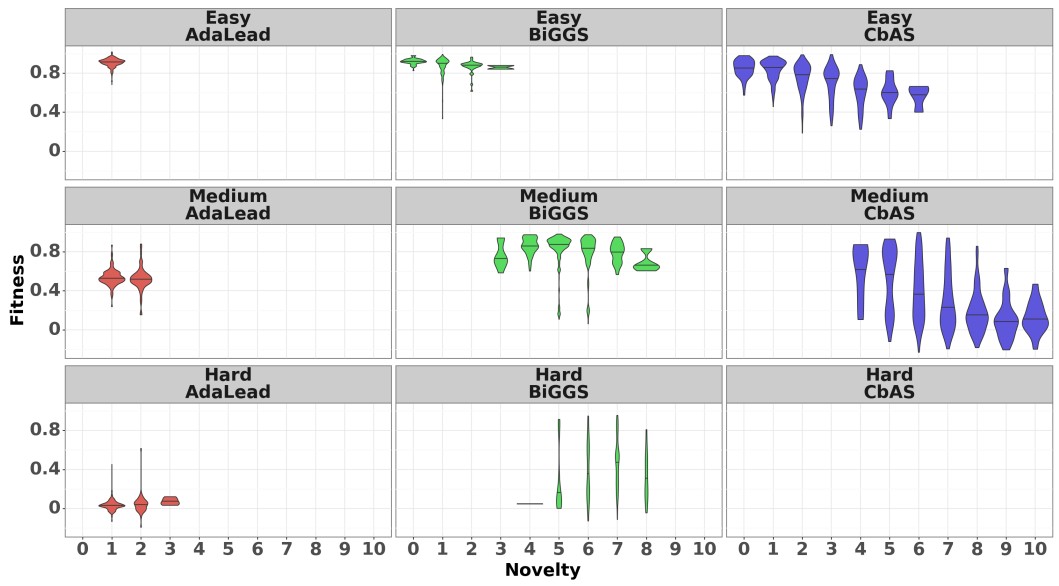

Figure 2: Comparison of GFP novelty and fitness on samples from AdaLead, BiGGS, and CbAS. From left to right, we observe increasing exploration behaviour from the respective methods. However, only BiGGS maintains high fitness while exploring the novel sequences. Nearly all samples from CbAS on hard are beyond 10 novelty and have very low fitness.

the other hand, manages to find the middle ground, displaying high fitness in its samples while also effectively exploring across the mutational gap at each difficulty level.

## 5.2 Ablations

We perform ablations on each component of BiGGS on the hard difficulty task. In the first ablation, we replace BiG with GWG but use an equivalent number of samples by running $R = 15$ of IE for a fair comparison. The second ablation removes GS and starts sampling after initial predictor training. The last ablation removes iterative extrapolation by setting $R = 1$, $M = 15$, $N_{\text{sample}} = 300$ which maintains the number of samples but without iterative rounds. Our results are shown in Table 3. We

Table 3: Ablation results (our oracle).

| Difficulty | Metric | **BiGGS** | with GWG | without IE | without GS |
|---|---|---|---|---|---|
| | Fitness | **0.43 (0.0)** | 0.38 (0.0) | 0.21 (0.0) | 0.0 (0.0) |
| Hard | Diversity | 4.1 (0.1) | 4.0 (0.1) | 8.3 (0.1) | 18.4 (0.6) |
| | Novelty | 7.0 (0.0) | 7.1 (0.2) | 4.0 (0.0) | 6.0 (0.0) |

see GS is crucial for BiGGS on the hard difficulty level. Additional analysis is provided in section 5.3. Removing IE also results in a large decrease in performance. Unsurprisingly, GWG greatly benefits from GS and IE due to its similarity with BiG. However, using BiG results in improved fitness. We conclude each component of BiGGS contributes to its performance.

## 5.3 Analysis

We analyze BiGGS in the hard GFP task and demonstrate that (1) GS results in gradients from BiG that point towards higher fitness sequences and (2) BiG's ability to sample large mutations ($M \geq 3$) enables efficient traversal of large mutational distances in a high dimensional space.

Figure 3A, B shows how GS leads to a smooth fitness landscape, enabling BiG to sample high-fitness mutations. Often but not always, GS allows BiGGS to assign high probability to higher-fitness mutations that are low-probability without GS. We use the GFP wildtype (WT) as a representative of high-fitness sequences in the 99th percentile. The smoothed $f_\theta$ (fig. 3A) assigns high probability to the mutation that changes the current residue to the WT residue at a given proposed position, while

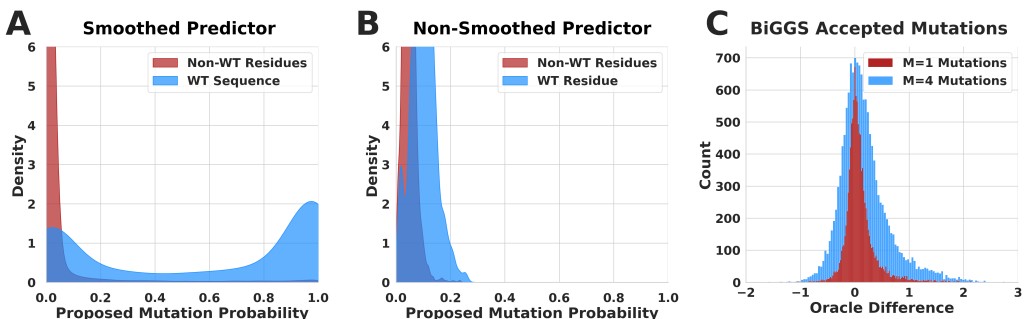

Figure 3: Analysis of BiGGS for Hard Task. **(A, B)** Proposed mutation probability of WT residue vs. non-WT residues for subsequently accepted mutations with and without GS. The non-smoothed predictor gives the WT residue only slightly higher probability than other residues. **(C)** Single vs. quadruple mutations accepted by BiGGS. Quadruple mutations lead to more large improvements.

giving low probability to other (lower fitness) mutations. The non-smoothed predictor proposes to mutate the current residue to the WT residue only slightly more often than other mutations (fig. 3B).

In Figure 3C, we show that BiGGS's ability to consider large mutations ($M \geq 3$) facilitates efficient exploration. We use the oracle to analyze all single ($M = 1$) and quadruple ($M = 4$) mutations accepted during the course of running BiGGS. We choose $M = 4$ as it represents the largest portion of BiGGS-accepted mutations among large mutations ($M \geq 3$). According to the oracle, the largest quadruple mutation fitness increases are bigger than the largest single mutation fitness increases. Quadruple mutations also result in a greater number of substantial fitness increases. We note a somewhat larger count of substantially negative mutations for quadruple mutations vs. for single mutations. This is expected given BiGGS's stochasticity, and the tendency of large mutations to be more deleterious than small ones. Similar analysis for $M$ up to 5 is in Appendix D.

## 6   Discussion

In this work, we presented BiGGS, a method for optimizing protein fitness by incorporating ideas from MCMC, graph Laplacian regularization, and directed evolution. We outlined a new benchmark on GFP that introduces the challenge of starting with poor-fitness sequences, many edits from the top fitness sequences. BiGGS discovered higher fitness sequences than in the starting set, even in the hard difficulty of our benchmark where prior methods struggled. We analyzed the two methodological advancements, Graph-based Smoothing (GS) and Bi-level Gibbs (BiG) (which includes Iterative Extrapolation), as well as ablations to conclude each of these techniques aided BiGGS's performance.

There are multiple extensions of BiGGS. The first is to improve BiG by removing the independence assumption across residues and instead modeling joint probabilities of epistatic interactions. One possibility for learning epistatic interactions is to incorporate 3D structure information (if available) to bias the sampling distribution. Secondly, the effectiveness of GS in our ablations warrants additional exploration into better regularization techniques for protein fitness predictors. Our formulation of GS is slow due to the nearest neighbor graph construction and its $L_1$ optimization. Lastly, investigating BiGGS to handle variable length sequences, multiple objectives, and multiple rounds of optimization is of high importance towards real protein engineering problems. Our code is included in the supplementary data and will be publicly available upon acceptance.

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
