# OpenReview forum: "Optimizing protein fitness using Bi-level Gibbs sampling with Graph-based Smoothing"
_NeurIPS.cc/2023/Conference — Submitted to NeurIPS 2023_

### Official Review · Reviewer_bM4W · 2023-07-05

**Soundness:** 3 good
**Presentation:** 3 good
**Contribution:** 2 fair
**Rating:** 5
**Confidence:** 2

**Summary:**

This paper proposes a sampling method for directed protein evolution. The method includes three parts: Bi-level Gibbs sampling (BiG), graph-based smoothing (GS), and iterative extrapolation (IE). BiG improves sampling efficiency for multi-point mutant sequences, GS helps to sample toward high fitness sequences, and IE reduces the Taylor approximation error. Experiments results show that the proposed method could achieve competitive performance.

**Strengths:**

- The presentation is clear and the idea is reasonable. The figures are beautiful and The authors clearly formalize the problem and key operations.
- The addressed problem is important. The relevant methods may be applied in different protein design scenarios.
- The experiments are convincing. I appreciate the authors' efforts in revealing the effect of each module via ablation studies.

**Weaknesses:**

- The proposed approach is somewhat heuristic and lacks theoretical assurance.
- There is a lack of relevant prior research on protein design [1-7].
- The proposed method does not compare with strong baselines such as the diffusion-based method [6,7]. Therefore, it is challenging for reviewers to verify its effectiveness and potential.

[1] Gao, Zhangyang, Cheng Tan, and Stan Z. Li. "Alphadesign: A graph protein design method and benchmark on alphafolddb." arXiv preprint arXiv:2202.01079 (2022).

[2] Hsu, Chloe, et al. "Learning inverse folding from millions of predicted structures." International Conference on Machine Learning. PMLR, 2022.

[2] Gao, Zhangyang, Cheng Tan, and Stan Z. Li. "PiFold: Toward effective and efficient protein inverse folding." The Eleventh International Conference on Learning Representations. 2022.

[3] Ingraham, John, et al. "Generative models for graph-based protein design." Advances in neural information processing systems 32 (2019).

[4] Tan, Cheng, et al. "Global-Context Aware Generative Protein Design." ICASSP 2023-2023 IEEE International Conference on Acoustics, Speech and Signal Processing (ICASSP). IEEE, 2023.

[5] Jing, Bowen, et al. "Learning from protein structure with geometric vector perceptrons." arXiv preprint arXiv:2009.01411 (2020).

[6] Stanton, Samuel, et al. "Accelerating bayesian optimization for biological sequence design with denoising autoencoders." International Conference on Machine Learning. PMLR, 2022.

[7] Gruver, Nate, et al. "Protein Design with Guided Discrete Diffusion." arXiv preprint arXiv:2305.20009 (2023).

**Questions:**

- Q1: How do you define $\text{dist}(v_i, v_j)$ when constructing the graph? Does the distance function consider the fitness score?

- Q2: When constructing the graph, are the sequences mutated randomly, or is there some guiding knowledge used to direct the mutations?

- Q3: Could you compare the proposed method with deep learning ones? Is it because there is a lack of work related to deep learning in this area?

---

> ### Author Rebuttal · Authors · 2023-08-09
>
> We thank the reviewer for their time and insightful feedback. Below we address the reviewer’s concerns and questions.
>
> > The proposed approach is somewhat heuristic and lacks theoretical assurance.
>
> Our approach is well-grounded in the theoretical guarantees of Gibbs With Gradients (GWG) [1]. Theorem 1 of [1] proved faster convergence of the sampling with GWG when the energy function (the predictor in our case) is smooth. Our Graph-based Smoothing (GS) is principled where our regularization objective (equation 4) follows the same regularization objective in spectral graph theory [2] and discrete regularization [3]. We show that GS achieves a reduction in the regularization objective. Hence, GWG and GS are complementary from their theoretical properties. The focus of our work is empirical but we are keen to explore additional theoretical properties in follow-up works.
>
> On heuristics, we agree our method has several hyperparameters that should be tuned on a problem by problem basis. This limitation is stated in lines 221-222 and is left as a follow-up work. Our technical contribution is a demonstration of successful (and state-of-the-art) use of GWG and GS.
>
> > There is a lack of relevant prior research on protein design [1-7].
>
> References [1-5] are on a different problem, inverse protein folding, where the task is to generate sequences that are most likely to fold into a given 3D structure. Our work is based on generating sequences that maximize a desired experimental fitness function such as fluorescence, gene delivery, etc. Structure is related to these properties however our problem formulation specifically does not assume access to the structure. We address references [6-7] in the next comment.
>
> > The proposed method does not compare with strong baselines such as the diffusion-based method [6,7]. Therefore, it is challenging for reviewers to verify its effectiveness and potential.
>
> Only [7] is a diffusion based approach and we note it is concurrent work that was preprint on May 31, well after the NeurIPS submission deadline. [6] is based on multi-objective optimization and does not apply its method to the benchmarks in our work. It is unclear where its performance stands in relation to our existing baselines when their method was not developed for single property optimization. We believe our current baselines represent state-of-the-art and allows for measuring our method’s effectiveness and potential.
>
> > Q1: How do you define dist(��,��) when constructing the graph? Does the distance function consider the fitness score?
>
> We define dist in line 232-233 which is the [Levenshtein distance](https://en.wikipedia.org/wiki/Levenshtein_distance) . We have not explored alternative distance functions since Levenshtein is used in all existing works and is standard for discrete data.
>
> > Q2: When constructing the graph, are the sequences mutated randomly, or is there some guiding knowledge used to direct the mutations?
>
> We mutate sequences randomly since we do not assume auxiliary information about the sequence. This way we can estimate the local fitness landscape around each starting sequence and estimate the graph Laplacian in a unbiased way.
>
> > Q3: Could you compare the proposed method with deep learning ones? Is it because there is a lack of work related to deep learning in this area?
>
> Our work is a deep learning approach since we train a neural network for the predictor, please see lines 81-83.
>
> [1] Grathwohl, Will, et al. "Oops i took a gradient: Scalable sampling for discrete distributions." International Conference on Machine Learning. PMLR, 2021.
>
> [2] Zhou, Dengyong, et al. "Learning with local and global consistency." Advances in neural information processing systems 16 (2003).
>
> [3] Zhou, Dengyong, and Bernhard Schölkopf. "Regularization on discrete spaces." Joint Pattern Recognition Symposium. Berlin, Heidelberg: Springer Berlin Heidelberg, 2005.

---

> > ### Comment · Reviewer_bM4W · 2023-08-16
> > **Reply to the rebuttal**
> >
> > Thanks for the authors' rebuttal.  After reading the response, I have other concerns:
> >
> > - I agree with the reviewer FEBk that the evaluation is based on machine learning models as oracles. Even though the oracle models are trained on experimental data, the fitness landscape in the oracle models might have very different properties compared to real protein fitness landscapes.
> >
> > - I still recommend authors to compare with deep-learning-based optimizing methods, such as diffusion models. I'm not sure whether the baselines are competitive.
> >
> > Considering the author's effort for clarification. I would like to raise the score from 4 to 5, but reduce the confidence.

---

> > > ### Author Response · Authors · 2023-08-16
> > > **Author response**
> > >
> > > Thank you for the response and increase in score!
> > >
> > > > I agree with the reviewer FEBk that the evaluation is based on machine learning models as oracles. Even though the oracle models are trained on experimental data, the fitness landscape in the oracle models might have very different properties compared to real protein fitness landscapes.
> > >
> > > We agree this is a limitation. However, **it is a limitation of many published machine learning (ML) protein engineering papers**. See our global comment of references [1-5] that were published in ML conferences and only used ML-based oracles. We hope it is apparent the limitation is not unique to our work.
> > >
> > > To help evaluate the technical merits of our method, we included a synthetic experiment (see global comment) where we know the fitness landscape and can evaluate our method’s performance with a true oracle. We demonstrate the effect of graph-based smoothing where the smoothed predictor is able to model the global upward slope of fitness while smoothing over the real local drops in fitness (see figure 2 of the rebuttal PDF). Our gradient-based sampling algorithm can exploit this smoothing to make progress towards the highest fitness sequences.
> > >
> > > > I still recommend authors to compare with deep-learning-based optimizing methods, such as diffusion models. I'm not sure whether the baselines are competitive.
> > >
> > > We understand the concern. To the best of our knowledge, **the only related diffusion work for protein fitness optimization is [1] which was put on arxiv May 31, 2023 — 2 weeks after our submission deadline**. We consider this as concurrent work, as stated in the [NeurIPS policy](https://neurips.cc/Conferences/2023/PaperInformation/NeurIPS-FAQ). We acknowledge diffusion is a promising method and are happy to include a discussion in the updated manuscript.
> > >
> > > **Brief discussion**: diffusion is not state-of-the-art (SOTA) in discrete sampling tasks [2,3]. The research moves rapidly and this may change soon. Our contribution, graph-based smoothing, is also applicable to discrete diffusion in potentially guiding a smoothed classifier to generate higher fitness sequences. We acknowledge this as an interesting future direction, possibly when discrete diffusion becomes more established.
> > >
> > > Lastly, we would like to clarify that **our baselines all use deep learning and encapsulate SOTA in protein fitness optimization with deep learning**. At time of submission, PEX [4] (included in our experiments) was the latest SOTA according to Google Scholar and had available code. A follow-up work [5] had a pre-print on January 3, 2023 but code was not available until ICML 2023. We believe our baselines are competitive and accurate representation of SOTA within the scope of our work, but welcome any feedback.
> > >
> > > [1] Gruver, Nate, et al. "Protein Design with Guided Discrete Diffusion." arXiv preprint arXiv:2305.20009 (2023).
> > >
> > > [2] Diffusion language models https://sander.ai/2023/01/09/diffusion-language.html
> > >
> > > [3] Austin, Jacob, et al. "Structured denoising diffusion models in discrete state-spaces." *Advances in Neural Information Processing Systems* 34 (2021): 17981-17993.
> > >
> > > [4] Ren, Zhizhou, et al. "Proximal exploration for model-guided protein sequence design." *International Conference on Machine Learning*. PMLR, 2022.
> > >
> > > [5] Chen, Can, et al. "Bidirectional learning for offline model-based biological sequence design." *arXiv preprint arXiv:2301.02931* (2023).

---

### Official Review · Reviewer_XaWK · 2023-07-08

**Soundness:** 3 good
**Presentation:** 3 good
**Contribution:** 3 good
**Rating:** 5
**Confidence:** 4

**Summary:**

This work applies gibbs sampling methods for generating random sequences of proteins to find mutants with high fitness. The model is compared against a set of sampling methods and the results are evaluated on GFP by fitness, diversity, and novelty.

**Strengths:**

- The problem formulation and model construction are presented clearly.
- The experiments were evaluated comprehensively from different perspectives.

**Weaknesses:**

- The motivation for random sampling the whole sequence for a protein in the context of is unclear.
- As AAV has been mentioned more than once in Introduction, it is preferred to put them in the main text.
- While the proposed solution to the problem is a sampling method, other baseline methods that have been used to solve the same problem might also be compared, such as the ESM-series.

**Questions:**

- (line 37) "to reach high-fitness sequences requiring many mutations..." is not rigorous. It is not true that modifying more sites will always return a better mutant.
- (line 49) GFP (in [30]) only has 238 amino acids, which is not "one of the longest proteins".
- (line 86) “Sampling random sequences and assigning them the lowest possible fitness value” might be risky, as it is hard to tell if a random sequence will actually lose its functionality. Also, at least for the GFP dataset, there is already a large portion of negative samples (i.e., mutants that perform worse than the wild-type), why it is essential to add more negative samples?
- (line 240) If diversity and novelty do not have a preferred direction or optimal value to reach, why are they reported in the tables and how should one make an analysis from them?

**Limitations:**

The authors did not analyze the potential negative societal impact of their work.

---

> ### Author Rebuttal · Authors · 2023-08-09
>
> We thank the reviewer for their time and insightful feedback. Below we address the reviewer’s concerns and questions.
> > The motivation for random sampling the whole sequence for a protein in the context of is unclear.
>
> Could the reviewer clarify which component of our method they refer to? Our method does not sample the entire sequence. It starts with a seed sequence then samples point mutations proportionally to the gradients of the smoothed predictor (equation 2). In graph-based smoothing, we also start with “parent” sequences in the training set and introduce random mutations to construct a graph. For negative data augmentation, we sample random sequences under the assumption they have zero fitness. This assumption is valid from many studies about the sparsity of protein fitness landscapes [1].
> > As AAV has been mentioned more than once in Introduction, it is preferred to put them in the main text.
>
> We are limited by space which placed the AAV result in the maintext. However, we will attempt to add if accepted with the extra page.  The inclusion of AAV in our work is to demonstrate our method is not over-optimized for GFP. Our method is able to achieve state-of-the-art results on both landscapes using our rigorous splits.
> > While the proposed solution to the problem is a sampling method, other baseline methods that have been used to solve the same problem might also be compared, such as the ESM-series.
>
> We include a comprehensive set of six other baselines that cover the range of methods published on protein fitness optimization: GFlowNets, VAEs, greedy search, bayesian optimization, conservative model-based optimization, and evolutionary search. The ESM-series do not include a method for optimizing fitness. The closest ESM model is ESM-IF [6] which is the task of designing sequences to fold into a structure. The only work we know of using ESM for fitness optimization are [3,4] which we cite but neither had open source implementations at time of submission.
> > (line 37) "to reach high-fitness sequences requiring many mutations..." is not rigorous. It is not true that modifying more sites will always return a better mutant.
>
> Indeed, this is not always true. However, if only a few mutations are required then using off the shelf techniques such as directed evolution would be highly effective. Our interest is in the scenario where many mutations are required to reach high fitness. We have updated the sentence to reflect this, “In cases where many mutations are required to reach high-fitness sequences…”
> > (line 49) GFP (in [30]) only has 238 amino acids, which is not "one of the longest proteins".
>
> Thank you for catching this. We have clarified this to state GFP is representative of a intractable search space that is infeasible to search over, “we primarily study GFP because of (1) its difficulty with 238 mutable positions (a 20^238 search space)…”
> > (line 86) “Sampling random sequences and assigning them the lowest possible fitness value” might be risky, as it is hard to tell if a random sequence will actually lose its functionality. Also, at least for the GFP dataset, there is already a large portion of negative samples (i.e., mutants that perform worse than the wild-type), why it is essential to add more negative samples?
>
> It has been well studied that fitness is very sparse and random sequences are highly likely to not be functional [1]. Following this, it is valid to assume random sequences will not have any fitness in particular for the desired fitness. We require additional negative samples since we set-up our benchmark such that only a small fraction of the data is available, i.e. the hard setting only allows for 1600 data points. This includes only a small number of true negative examples so negative data augmentation is valuable.
> > (line 240) If diversity and novelty do not have a preferred direction or optimal value to reach, why are they reported in the tables and how should one make an analysis from them?
>
> Reporting diversity and novelty was started with GFlowNets [5] which has since then become standard practice for comparing protein design methods. These metrics are important to measure coverage of sequence space (diversity) while generalizing beyond the training set (novelty). But high diversity and novelty does not matter if the median fitness of the sequences is always very low which we found to happen in several baselines (table 2). Diversity and novelty can also be used a tiebreaker: if two methods achieve the same high fitness then the method with the higher diversity or novelty should be preferable.
> > The authors did not analyze the potential negative societal impact of their work.
>
> Thank you for pointing out this accidental omission! We have added the following line into our discussion. “Protein engineering is an important problem with high impact in improving medicine and biotechnology. However, it can also lead to discovery of harmful proteins such as viruses. While the method is not yet mature, mitigating harmful uses is an important consideration.”
>
> [1] Brookes, David H., Amirali Aghazadeh, and Jennifer Listgarten. "On the sparsity of fitness functions and implications for learning." Proceedings of the National Academy of Sciences 119.1 (2022): e2109649118.
>
> [2] Sinai, Sam, et al. "AdaLead: A simple and robust adaptive greedy search algorithm for sequence design." arXiv preprint arXiv:2010.02141 (2020).
>
> [3] Lee, Minji, et al. "Protein sequence design in a latent space via model-based reinforcement learning." (2022).
>
> [4] Emami, Patrick, et al. "Plug & play directed evolution of proteins with gradient-based discrete MCMC." Machine Learning: Science and Technology 4.2 (2023): 025014.
>
> [5] Jain, Moksh, et al. "Biological sequence design with gflownets." International Conference on Machine Learning. PMLR, 2022.
>
> [6] Hsu, Chloe, et al. "Learning inverse folding from millions of predicted structures." International Conference on Machine Learning. PMLR, 2022.

---

> > ### Comment · Reviewer_XaWK · 2023-08-17
> >
> > Thanks for addressing the questions I asked before. "The motivation of random sampling..." was related to the content in line 85 "we double the size of the dataset by sampling random sequences". I can understand why the authors designed this augmentation after reading their reply., but I still feel this operation is fairly tricky since there are already tons of negative samples, and randomly sampling the whole sequence does not only introduce unfunctional proteins, in most cases, they might just be meaningless phrases. After reading the authors' reply as well as other reviewers' comments, I would like to maintain my initial score.

---

> > > ### Author Response · Authors · 2023-08-17
> > > **Author response**
> > >
> > > Thank you for reading the rebuttal and responses. We would like to clarify a possible confusion.
> > >
> > > > I still feel this operation is fairly tricky since there are already tons of negative samples
> > > >
> > >
> > > **While the whole dataset contains many negative samples, we purposely only train the model on a curated subset to stimulate the realistic setting where the initial training set is very small.** We copy our table 1 below with the dataset statistics of the medium and hard difficulties,
> > >
> > > ```
> > > | Difficulty | Fitness range | Dataset size | Gap |
> > > | ---------- | ------------- | ------------ | --- |
> > > | Medium     | 20-40th       | 2828         | 6   |
> > > | Hard       | 10-30th       | 1636         | 7   |
> > > ```
> > >
> > > These difficulties limit the starting set to low fitness with around 1,600 sequences in the Hard setting.  The model benefits from random negative examples that we know have no fitness. Otherwise random sequences can become adversarial examples the model may spuriously predict to have high fitness.
> > >
> > > Our negative data augmentation is also simple: double the dataset size by randomly sampling sequences and assigning lowest fitness. We copy lines 83-86 here,
> > >
> > > > We found it beneficial to employ negative data augmentation… we double the size of the dataset by sampling random sequences and assigning them the lowest possible fitness value.
> > > >
> > >
> > > Note that negative augmentation is more complicated in other domains such as vision [1, 2] where random rotations, blurring, and GANs are employed. We believe our operation is easy for a simple improvement in our predictor performance. Note our improved predictor is shared across our model-based optimization baselines.
> > >
> > > > randomly sampling the whole sequence does not only introduce unfunctional proteins, in most cases, they might just be meaningless phrases.
> > > >
> > >
> > > Agreed. They are "meaningless phrases" that are obvious to us; however, the model does not know this a priori and may assign high fitness to examples it has not seen before. For instance, the seminal work Ilya et al [3] showed deep learning models that can incorrectly classify pure noise as frogs.
> > >
> > > We hope this clarifies any confusions or concerns regarding the method. Regardless of the score, we are appreciative of your interaction and time!
> > >
> > > [1] https://arxiv.org/pdf/2102.05113.pdf
> > >
> > > [2] https://arxiv.org/pdf/1706.06083.pdf
> > >
> > > [3] https://arxiv.org/abs/1905.02175

---

### Official Review · Reviewer_edmB · 2023-07-10

**Soundness:** 3 good
**Presentation:** 2 fair
**Contribution:** 3 good
**Rating:** 5
**Confidence:** 3

**Summary:**

This paper proposes a novel method to optimize protein fitness by using bi-level Gibbs sampling with graph-based smoothing. Experiments on real datasets are used to verify the effectiveness of the proposed method.

**Strengths:**

1. The studied problem about designing novel protein is interesting and challenging.

2. The proposed method seems to be novel.

3. Experiments show that the proposed method can outperform other baselines to achieve the state-of-the-art performance.


**Weaknesses:**

The writing can be improved. There exist many typos and grammatical errors. Line 106, where is Algorithm 2? Furthermore, the organization can also be improved. For example, Section 3 can be integrated into “Experiment”. It is better to move Section 4 to be just after “Introduction” or just before “Conclusion”.

**Questions:**

In Table 3, the fitness is zero without GS. Can you give more explanation about this? Does it mean that other parts of the proposed method have no ability to model the problem?

**Limitations:**

yes

---

> ### Author Rebuttal · Authors · 2023-08-09
>
> We thank the reviewer for their time and insightful feedback. Below we address the reviewer’s concerns and questions.
>
> > The writing can be improved. There exist many typos and grammatical errors. Line 106, where is Algorithm 2? Furthermore, the organization can also be improved. For example, Section 3 can be integrated into “Experiment”. It is better to move Section 4 to be just after “Introduction” or just before “Conclusion”.
>
> Thank you for the suggestions. We have done a sweep and corrected typos and grammatical errors. Due to space limitations, we had to put algorithm 2 in the appendix. We agree with the re-organization of the sections and have moved Section 4 (Related works) to be after the introduction. We have kept Section 3 (Benchmarks) as a separate section because we believe it is an important contribution on its own. Section 3 now precedes the “Experiment” section.
>
> > In Table 3, the fitness is zero without GS. Can you give more explanation about this? Does it mean that other parts of the proposed method have no ability to model the problem?
>
> Table 3 shows the importance of Graph-based Smoothing (GS) when using Gibbs With Gradients (GWG) [1]. GWG was demonstrated to be state-of-the-art in many discrete optimization and sampling problems but its theoretical guarantees (Theorem 1 of [1]) assumes a smooth energy function. The energy function in our case is the fitness predictor (see Section 3.2 of our work). Table 3 reflects the theory of GWG in that a smoothed fitness predictor is necessary but the pay-off is excellent performance in discrete optimization. **It is the synergy of smoothing and GWG that achieves our key result.** Lastly, we note prior baselines also achieve a fitness close to 0 on the hard setting which remarkably puts our method in a class of its own when the fitness landscape is very sparse and rugged.
>
> [1] Grathwohl, Will, et al. "Oops i took a gradient: Scalable sampling for discrete distributions." International Conference on Machine Learning. PMLR, 2021.

---

> > ### Comment · Reviewer_edmB · 2023-08-20
> > **Thanks for the response**
> >
> > Thank you for the response.
> >
> > Does it mean that other parts of the proposed method have no ability to model the problem?
> >
> > Please directly reply to the question. The fact that prior baselines also achieve a fitness close to 0 does not mean that it is reasonable for other parts of the proposed method to have no ability to model the problem.

---

> > > ### Author Response · Authors · 2023-08-20
> > > **Author response**
> > >
> > > > Does it mean that other parts of the proposed method have no ability to model the problem?
> > > >
> > >
> > > Our ablations (table 3) show the sampling technique Gibbs with Gradients (GWG) is unable to model the problem without graph-based smoothing (GS). Note GWG is not developed by us. Our insight is the combination of GWG with our proposed regularization, GS, leads to its improved performance.
> > >
> > > Lastly, we propose Iterative Extrapolation (IE) as a simple procedure to resample with GWG while controlling computation cost through clustering. This again caters to GWG, which relies on a Taylor approximation of the proposal distribution, equation (1), that requires recomputing gradients after each sampling step. Our ablations show IE is not necessary to get non-zero performance but leads to a large gain.
> > >
> > > > The fact that prior baselines also achieve a fitness close to 0 does not mean that it is reasonable for other parts of the proposed method to have no ability to model the problem.
> > > >
> > >
> > > Thank you for the concern. Could the reviewer elaborate why it is not reasonable? Since the discussion period ends tomorrow, we will try to address the concern with our interpretation.
> > >
> > > The concern may be that other parts of our method are not needed to achieve the high fitness we report in the hard difficulty. We accounted for this by using GS for all our model-based optimization (MBO) baselines, see line 218-219 — also copied here: “To ensure a fair comparison, we use the same predictor across all model-based baselines.” This includes GFN-AL, AdaLead, CoMs, and PEX. Our results (table 3) show none of these methods can utilize the smoothness of the predictor in sampling higher fitness.
> > >
> > > These results agree with the theory of GWG, which assumes the smoothness of the predictor when sampling proportionally to the gradients towards higher fitness. Therefore, we believe it is reasonable that GWG fails when the predictor is not smooth but performs exceptionally when we satisfy its underlying assumption.

---

### Official Review · Reviewer_FEBk · 2023-07-25

**Soundness:** 3 good
**Presentation:** 3 good
**Contribution:** 2 fair
**Rating:** 4
**Confidence:** 4

**Summary:**

This work considers the problem of optimization over a noisy fitness landscape of sequences. There are two main contributions: 1) Graph-based smoothing (GS) --- considering the landscape as a noisy landscape and applying L1 graph Laplacian regularization. 2) Bi-level Gibbs (BiG) --- using bi-level sampling to propose 5 indices to mutate simultaneously.

To evaluate the performance of the proposed method, the authors used design benchmarks based on oracles trained on the GFP and AAV datasets, and showed improved sequence optimization on the synthetic design benchmarks.

**Strengths:**

This work proposes a new sampling method (Bi-level Gibbs with Graph-based Smoothing) in the protein sequence optimization problem. The proposed method is well-motivated. In particular, graph-based smoothing is an interesting approach to address the non-smoothness of protein fitness landscapes. The authors gave a clear description of the proposed method and presented detailed empirical results on simulated data based on real-world experimental data. In the evaluation, the authors took care to avoid overfitting the algorithm and included two datasets (GFP and AAV) to study two different protein fitness landscapes with different properties. The ideas in this work could inspire future work around this important topic.

**Weaknesses:**

The main weakness is in the empirical evaluation. The evaluation is based on machine learning models as oracles. Even though the oracle models are trained on experimental data, the fitness landscape in the oracle models might have very different properties compared to real protein fitness landscapes. For example, for graph-based smoothing to work well, it seems like some amount of smoothness in the landscape is needed, and machine learning models might present artificially smoother landscapes than real fitness landscapes. Therefore, one major concern is that the empirical results are tied to the oracle models.

That being said, it is notoriously challenging to evaluate protein sequence design algorithms as doing so rigorously would require collecting new data points, so this is not a unique weakness to this particular work at hand. In short of performing experimental evaluation, it could relatively strengthen the paper to study more fitness landscapes in the literature and demonstrate evidence that they do have the similar characteristics as the oracle landscapes.

**Questions:**

See "weaknesses" and "limitations".

**Limitations:**

The discussion section does mention limitations around handling variable length sequences and computational speed. It would also be helpful to include a detailed discussion on what types of fitness landscapes BiG would be well-suited for, and counter-examples of fitness landscapes where BiG fails to find optimal solutions. In particular, is it always a good idea to mutate many positions simultaneously? are there situations where it would be preferable to do one mutation at a time?

---

> ### Author Rebuttal · Authors · 2023-08-09
>
> We thank the reviewer for their time and insightful feedback. Below we address the reviewer’s concerns and questions.
>
> > The main weakness is in the empirical evaluation. The evaluation is based on machine learning models as oracles…. it is notoriously challenging to evaluate protein sequence design algorithms as doing so rigorously would require collecting new data points, so this is not a unique weakness to this particular work at hand. In short of performing experimental evaluation, it could relatively strengthen the paper to study more fitness landscapes in the literature and demonstrate evidence that they do have the similar characteristics as the oracle landscapes.
>
> We agree experimental validation is a limitation of most method papers — including ours — on protein fitness optimization. As discussed in the global comment, we hope the same standards of previously published works can be applied to us.
>
> We argue our results are more rigorous than previous works. Our experiments study three splits of varying difficulty over GFP and AAV while previous works only studied one split (see section 3). For Green Fluorescence Protein (GFP), we discovered the previous split contained data leakage of the highest fitness sequences. Our corrected splits show prior methods perform poorly in generalizing beyond the training set. We include source code for transparency and demonstrate a need for quality over quantity of protein engineering benchmarks. Analyzing GFP and AAV to construct careful splits takes time which we argue is more important than evaluating many datasets without in-depth analysis of each.
>
> > Even though the oracle models are trained on experimental data, the fitness landscape in the oracle models might have very different properties compared to real protein fitness landscapes. For example, for graph-based smoothing to work well, it seems like some amount of smoothness in the landscape is needed, and machine learning models might present artificially smoother landscapes than real fitness landscapes.
>
> Reviewer DKAZ had similar concerns. Indeed, our method deviates from the realistic, non-smooth protein landscape and replaces it with a smoothed proxy. We argue this is desirable because we wish to ignore sharp, local optimas in order to only follow gradients that guide to the highest fitness sequences in our gradient-based sampling method. Since protein fitness landscapes are difficult to analyze, we have included a synthetic fitness landscape (see global comment) that is un-smooth by construction but graph-based smoothing creates a smoothed proxy. We see smoothing improves the performance of our method despite it deviating from the true landscape. Furthermore, our ablations in table 3 and figure 3 demonstrate the importance of smoothing to achieve state-of-the-art results. We understand this is counterintuitive but regularization has been known to be an important factor in machine learning success.
>
> > Therefore, one major concern is that the empirical results are tied to the oracle models.
>
> We show in appendix figure 6 that our CNN oracle learns the GFP dataset more accurately than the previous Transformer oracle [3]. To show our results are not tied to an oracle, we included results using our CNN oracle (Table 2) and previous Transformer oracle (Table 5) both of which our method gets state-of-the-art results.
>
> > It would also be helpful to include a detailed discussion on what types of fitness landscapes BiG would be well-suited for, and counter-examples of fitness landscapes where BiG fails to find optimal solutions.
>
> We now include a case study with synthetic data as a fitness landscape that is rugged due to additive gaussian noise. We see smoothing improves the results with this particular type of landscape. Unfortunately, characterizing the GFP and AAV landscapes are difficult due to them being high dimensional.
>
> We have added the following discussion in section 6, “Our results indicate GGS (new name) is well-suited for our studied landscapes. Characterizing when it fails and succeeds is an important future direction. In particular, one could study the fitness landscape through fourier analysis as done in [5] or spectral graph theory [6] of the graph Laplacian we have introduced (section 3.3). Analyzing the relationship of smoothness in fitness landscapes and success of machine learning methods is of high interest as our results reveal.”
>
> > In particular, is it always a good idea to mutate many positions simultaneously? are there situations where it would be preferable to do one mutation at a time?
>
> Mutating many positions is intractable due to the exponential increase of possibilities. The benefit of our method is it enables performing iterative point mutations that traverse intermediate sequences of low fitness before arriving at a higher fitness sequence. This is possible due to the smoothing which places high probability on the intermediate sequences since they can lead to higher fitness. We shows this is the case for our synthetic case study in the global comment.
>
> [1] Sinai, Sam, et al. "AdaLead: A simple and robust adaptive greedy search algorithm for sequence design." arXiv preprint arXiv:2010.02141 (2020).
>
> [2] Ren, Zhizhou, et al. "Proximal exploration for model-guided protein sequence design." International Conference on Machine Learning. PMLR, 2022.
>
> [3] Trabucco, Brandon, et al. "Design-bench: Benchmarks for data-driven offline model-based optimization." International Conference on Machine Learning. PMLR, 2022.
>
> [4] Brookes, David H., Amirali Aghazadeh, and Jennifer Listgarten. "On the sparsity of fitness functions and implications for learning." Proceedings of the National Academy of Sciences 119.1 (2022): e2109649118.
>
> [5] Zhou, Dengyong, and Bernhard Schölkopf. "Regularization on discrete spaces." Joint Pattern Recognition Symposium. Berlin, Heidelberg: Springer Berlin Heidelberg, 2005.

---

> ### Comment · Area_Chair_GJjd · 2023-08-21
> **Acknowledge the rebuttal [ACTION REQUESTED]**
>
> Dear Reviewer FEBk,
>
> If you haven't already, please read the authors' rebuttal to your review and those of other reviewers. The authors have provided a reasonably detailed rebuttal. I kindly ask that you indicate whether your opinion of the paper has changed or if you require additional clarifications.
>
> Best regards,
> Your Area Chair

---

### Official Review · Reviewer_DKAZ · 2023-07-27

**Soundness:** 3 good
**Presentation:** 3 good
**Contribution:** 3 good
**Rating:** 4
**Confidence:** 3

**Summary:**

In this paper, the authors proposed a Gibbs sampling algorithm with graph signal smoothing that can generate high fitness protein sequence. The proposed method, BiGGS, constructs a smoothed space between protein sequence nodes, and use the gradient of a trained fitness predictor to guide the sampling of both new locations and its corresponding mutations. The authors also provided benchmark dataset with different level of difficulties to validate the efficacy of BiGGS, as well as another dataset that is of different fitness measure to showcase the generalization capability.

**Strengths:**

The problem of valid protein sequence generation is popularly studied in both computational biology and machine learning community. The proposed method is a combination of traditional sampling techniques, neural predictor for protein sequence, and graph signal smoothing. Overall the paper is well-written with minor typos that don't affect my reading. The problem is well-stated with each component of the method clearly described in details. The authors also find the issues with existing benchmark datasets, which are labelled as "easy setting" in this paper based on the distance between highest fitness sequence and the initial set of sequences, and proposed more challenge version of it. In the experimentation section, the authors provided extensive comparison with different baseline methods and BiGGS exhibits significant performance improvement. The ablation study also provide insights on which component brings the most improvement.

**Weaknesses:**

Besides the strengths mentioned above, there are several areas that can be improved:
- The hyperparameters seems to be chose arbitrary and the hyperparameter tuning seems difficult. The justification of choice is also missing.
- The complexity of method is quite high, especially the retraining step after smoothing, which could be a major bottleneck of optimizing the methods in other use cases.
- The intuition behind graph smoothing is somewhat contradicting to the "non-smooth" assumption in protein sequence fitness space, where changing one position will greatly shift the fitness score. Using single mutation neighbors are a bit worrying due to this reason.
- Lack of comparison with different predictors, or evidence of the existing oracle won't work with the framework.


**Questions:**

The proposed method seems like a greedy approach, yet it managed to find high fitness samples outside of the initial set. Is there any explanation on why the smoothing and gradient descent/ascend will help avoid local optima and reach higher value samples?

**Limitations:**

The authors adequately addressed the limitations and I don't a potential negative societal impact of this work.

---

> ### Author Rebuttal · Authors · 2023-08-09
>
> We thank the reviewer for their time and insightful feedback. Below we address the reviewer’s concerns and questions.
>
> > The complexity of method is quite high, especially the retraining step after smoothing, which could be a major bottleneck of optimizing the methods in other use cases.
>
> We address this in two ways. First, we discovered a simplification of sampling by removing the bi-level step which we found to not be necessary, see global comment. Removing this step still achieves state-of-the-art results and focuses our main contribution around graph-based smoothing (GS).
>
> Second, we believe simplification of GS is a follow-up study, but this relies on a working instance of GS. Our work demonstrates the first promising application of discrete regularization [1] in protein sequence optimization. Our study propose an promising direction of interdisciplinary research bridging regularization and graph theory with important biological applications. To promote usability, we have included documented code and benchmarks for practitioners and researchers.
>
> We emphasize our improved benchmarks in section 3 where in the hard difficulty the training set is small (~1000) and many mutations (greater than 5) are needed. **Other methods cannot come close to succeeding whereas ours is the only one to achieve good results** (see table 2). Even if a method is complex, we believe it is important to have working methods for new machine learning challenges regardless of complexity.
>
> > The hyperparameters seems to be chose arbitrary and the hyperparameter tuning seems difficult. The justification of choice is also missing.
>
> Our method has two types of hyperparameters. The first is based on how much compute is available such as the number of sampling steps, clusters in iterative extrapolation, and neighbors in graph-based smoothing. We chose reasonable values for our compute budget – our method runs within an hour. Most hyperparameter falls into this category.
>
> The second type of hyperparameter is problem dependent which for us are the sampling temperature and neural network hyperparameters. These hyperparameters require sweeps over multiple values and additional analysis. We performed standard tuning of neural network parameters hat best fit a held-out set. We found a temperature of 0.01 to work the best but have now included additional results (see below) in the appendix to show performance across temperatures. Note that temperature is a standard hyperparameter that needs to be tuned for sampling, i.e. Markov Chain Monte Carlo (MCMC), techniques and is not unique to our method.
>
> | Temperature (tau) | Fitnes | Diversity | Novelty |
> | ----------------- | ------ | --------- | ------- |
> | 1                 | 0      | 14.5      | 2.4     |
> | 0.5               | 0      | 14.1      | 5.4     |
> | 0.1               | 0.7    | 2.5       | 7       |
> | 0.01              | 0.81   | 2.6       | 7       |
>
> We state in line 221-222 that guidelines for hyperparameter selection is an important future direction. We have added the above discussion in the manuscript to justify our choices.
>
> > The intuition behind graph smoothing is somewhat contradicting to the "non-smooth" assumption in protein sequence fitness space, where changing one position will greatly shift the fitness score. Using single mutation neighbors are a bit worrying due to this reason.
>
> Indeed, our method deviates from the realistic, non-smooth protein landscape and replaces it with a smoothed proxy. We argue this is desirable because we wish to ignore sharp, local optimas in order to only following gradients that guide to the highest fitness sequences. Since protein fitness landscapes are difficult to analyze, we have included a synthetic fitness landscape (see global comment) where smoothing can be verified to be helpful even though it deviates from the true fitness landscape (as long the gradients point towards the peaks). Remarkably point mutations suffice to traverse smoothed landscapes. We show in our global comment that single mutations out performs performing multiple mutations.
>
> > Lack of comparison with different predictors, or evidence of the existing oracle won't work with the framework.
>
> We apologize this was not clear in our manuscript. Appendix figure 6 shows comparison of our CNN-based predictor with a commonly used Transformer predictor [2] in GFP fitness prediction. We find our predictor is more robust with less false positives. Evaluating multiple predictors/oracles is not the focus of our work, but we found it necessary to improve the existing oracle due to its unreliable performance.
>
> > The proposed method seems like a greedy approach, yet it managed to find high fitness samples outside of the initial set. Is there any explanation on why the smoothing and gradient descent/ascend will help avoid local optima and reach higher value samples?
>
> Our method is not greedy nor does it perform gradient ascent. We use a sampling algorithm where the distribution (equation 2) is constructed to allow for exploration while assigning higher probability to high fitness sequences based on our predictor. In particular, the temperature controls how greedy the distribution is while [Metropolis-Hastings](https://en.wikipedia.org/wiki/Metropolis–Hastings_algorithm) (equation 3) ensures the Markov chain achieves the desired distribution [3].
> As stated previously, smoothing the predictor allows for ignoring local optima while preserving information about higher fitness. We performed analysis on the effect of smoothing in Figures 3A and 3B where only the smoothed predictor exhibits gradients towards the highest fitness sequences during sampling.
>
> [1] Zhou, Dengyong, and Bernhard Schölkopf. "Regularization on discrete spaces." Joint Pattern Recognition Symposium. Berlin, Heidelberg: Springer Berlin Heidelberg, 2005.
>
> [2] Rao, Roshan, et al. "Evaluating protein transfer learning with TAPE." Advances in neural information processing systems 32 (2019)

---

> ### Comment · Area_Chair_GJjd · 2023-08-21
> **Acknowledge the rebuttal [ACTION REQUESTED]**
>
> Dear Reviewer DKAZ,
>
> If you haven't already, please read the authors' rebuttal to your review and those of other reviewers. The authors have provided a reasonably detailed rebuttal. I kindly ask that you indicate whether your opinion of the paper has changed or if you require additional clarifications.
>
> Best regards,
> Your Area Chair

---

### Official Review · Reviewer_65UE · 2023-07-27

**Soundness:** 3 good
**Presentation:** 3 good
**Contribution:** 3 good
**Rating:** 6
**Confidence:** 1

**Summary:**

The paper introduces BiGGS, a new sequence-based protein fitness optimization algorithm. BiGGS employs bi-level Gibbs sampling for efficient mutation sampling, graph-based smoothing to regularize the fitness landscape, and iterative extrapolation for progressive mutation towards higher fitness. The algorithm's effectiveness is demonstrated through benchmarks on Green Fluorescent Proteins and Adeno-Associated Virus, where BiGGS exhibits state-of-the-art performance in optimizing protein fitness.

**Strengths:**

1. The introduction of BiGGS, a novel sequence-based protein fitness optimization algorithm that incorporates ideas from MCMC, graph Laplacian regularization, and directed evolution.

2. The authors establish a novel benchmark on Green Fluorescent Proteins which presents the unique challenge of initiating the process with sequences of low fitness that require numerous edits to reach peak fitness.

3. The experimental results convincingly demonstrate that BiGGS surpasses other recently proposed methods in terms of both fitness and novelty, marking a significant advancement in the field.

4. The authors provide comprehensive details about the datasets, methods, and evaluation metrics used in their experiments. The well-structured and documented code is an added advantage.

5. The paper also discusses potential extensions of BiGGS, including improving BiG by removing the independence assumption across residues, exploring better regularization techniques for protein fitness predictors, and investigating BiGGS to handle variable length sequences, multiple objectives, and multiple rounds of optimization.

**Weaknesses:**

1. While the paper presents compelling results, it could be further enhanced by incorporating a case study that applies the proposed method in a real-world scenario. This would provide tangible evidence of the method's effectiveness in practical applications. For instance, the abstract asserts that "Our method is state-of-the-art in discovering high-fitness proteins with up to 8 mutations from the training set", but this claim lacks further substantiation within the body of the paper.

2. The paper's evaluation methodology heavily depends on a fitness score, calculated through a trained, black-box neural network. This approach raises questions about the results' reliability and interpretability. It would be beneficial if the authors could explore strategies to increase the transparency and comprehensibility of the evaluation process.


3. As someone unfamiliar with protein fitness optimization, I find it challenging to gauge whether this highly specialized task would be of interest to the broader NeurIPS community.

**Questions:**

Could the authors provide information on the prediction error of the trained oracle when applied to unseen sequences? How might this error impact the accuracy of the evaluation?


**Limitations:**

The paper does not explicitly address any limitations. However, from my viewpoint, a significant limitation lies in the disparity between the evaluation pipeline employed in the paper and the realities of practical application scenarios. The authors could strengthen their work by acknowledging and discussing this gap.

---

> ### Author Rebuttal · Authors · 2023-08-09
>
> We thank the reviewer for their time and insightful feedback. Below we address the reviewer’s concerns and questions.
>
> > While the paper presents compelling results, it could be further enhanced by incorporating a case study that applies the proposed method in a real-world scenario…
>
> We agree a case study is of high importance. However, not everyone has access to biological validation. In particular, previous protein sequence optimization methods in ML conferences have not used biological validation [1-6]. Allowing computational methods for validation fosters rapid exploration of methodology which biologists can then select from. We believe our evaluation is in line with current practices.
>
> An important contribution of our work is questioning the validity of previous benchmarks. We find data leakage was present in GFP and so the task was overly simplified. That is why in section 3 we introduce different difficulty splits for a rigorous benchmarking.
>
> > The abstract asserts that "Our method is state-of-the-art in discovering high-fitness proteins with up to 8 mutations from the training set", but this claim lacks further substantiation within the body of the paper.
>
> We have changed our claim to “Our method is state-of-the-art in discovering high-fitness proteins **based on a trained oracle** with up to 8 mutations from the training set” to clarify this dependence in using an oracle as the fitness score.
>
> > The paper's evaluation methodology heavily depends on a fitness score, calculated through a trained, black-box neural network. This approach raises questions about the results' reliability and interpretability. It would be beneficial if the authors could explore strategies to increase the transparency and comprehensibility of the evaluation process.
> Could the authors provide information on the prediction error of the trained oracle when applied to unseen sequences? How might this error impact the accuracy of the evaluation?
>
> Prior works evaluated using a fitness score calculated with a trained neural network [1-6]. We follow this practice as a proxy for real-world validation (see above). However, we found the previously used DesignBench (DB) oracle [6] to spurious correlations and low robustness as a fitness score when evlauated on held-out data. We developed a new oracle with higher robustness and higher spearman rho (see appendix figure 6). Prior works have not done this analysis of the oracle. For transparency, we included results with DB in appendix table 5. We achieve state-of-the-art results using both oracles.
>
> > As someone unfamiliar with protein fitness optimization, I find it challenging to gauge whether this highly specialized task would be of interest to the broader NeurIPS community.
>
> Protein fitness optimization would have a profound impact in advancing medicine [7,8]. We argue it is not a specialized task and has many technical unsolved questions [9] that are of high interest to the NeurIPS community such as active learning and discrete optimization. Machine learning has led to breakthroughs in biology such as AlphaFold2 [10] and RFdiffusion [11]. We hope this convinces the reviewer of this problem’s importance and relevance.
>
> > The paper does not explicitly address any limitations… a significant limitation lies in the disparity between the evaluation pipeline employed in the paper and the realities of practical application scenarios. The authors could strengthen their work by acknowledging and discussing this gap.
>
> Thank you for catching this. The last paragraph in Section 6 (Discussion) implies several limitations such as speed of graph-based smoothing and handling variable length sequences. We agree this is not explicit and thank the reviewer for pointing this out. It is updated now to say, “There are multiple **limitations** and extensions...”
>
> Regarding evaluation, we have added the following lines:
> * Section 3 (Benchmarks): “We note our results with trained oracles serve as a proxy for real-world experimental validation.”
> * Section 6 (Discussion): “Following prior works, we evaluate performance with trained oracles but emphasize these results serve as a proxy for experimental validation.”
>
> [1] Angermueller, Christof, et al. "Model-based reinforcement learning for biological sequence design." International conference on learning representations. 2019.
>
> [2] Brookes, David, Hahnbeom Park, and Jennifer Listgarten. "Conditioning by adaptive sampling for robust design." International conference on machine learning. PMLR, 2019.
>
> [3] Jain, Moksh, et al. "Biological sequence design with gflownets." International Conference on Machine Learning. PMLR, 2022.
>
> [4] Ren, Zhizhou, et al. "Proximal exploration for model-guided protein sequence design." International Conference on Machine Learning. PMLR, 2022.
>
> [5] Trabucco, Brandon, et al. "Conservative objective models for effective offline model-based optimization." International Conference on Machine Learning. PMLR, 2021.
>
> [6] Trabucco, Brandon, et al. "Design-bench: Benchmarks for data-driven offline model-based optimization." International Conference on Machine Learning. PMLR, 2022.
>
> [7] Arunachalam, Prabhu S., et al. "Adjuvanting a subunit COVID-19 vaccine to induce protective immunity." Nature 594.7862 (2021): 253-258.
>
> [8] Quijano-Rubio, Alfredo, et al. "The advent of de novo proteins for cancer immunotherapy." Current Opinion in Chemical Biology 56 (2020): 119-128.
>
> [9] Yang, Kevin K., Zachary Wu, and Frances H. Arnold. "Machine-learning-guided directed evolution for protein engineering." Nature methods 16.8 (2019): 687-694.
>
> [10] Jumper, John, et al. "Highly accurate protein structure prediction with AlphaFold." Nature 596.7873 (2021): 583-589.
>
> [11] Watson, Joseph L., et al. "De novo design of protein structure and function with RFdiffusion." Nature (2023): 1-3.

---

> > ### Comment · Reviewer_65UE · 2023-08-16
> > **Response**
> >
> > Thank you to the authors for your detailed response. After reading the response and other reviews, I've come to understand that most of the problems I mentioned are related to common weaknesses in this field and not specifically to this paper.
> >
> > However, I would like to clarify that by referring to 'a case study,' I'm not meaning biological validation. Rather, I mean the demonstration of how the model is performed on an example from the dataset (I guess Figure 1 is a synthetic example). Additionally, I'm still unclear on how you derived the number 8 in the phrase 'up to 8 mutations' within the abstract.
> >
> > As an emergency reviewer not well-versed in this field, I didn't identify any significant technical flaws in this paper. Consequently, I recommend acceptance, though my confidence in this recommendation is low.

---

> > > ### Author Response · Authors · 2023-08-16
> > > **Author response**
> > >
> > > Thank you for your reply! We are glad our response to clarified experimental validation concerns. Regarding the demonstration, we have provided a synthetic case study in our rebuttal of demonstrating the technical merits of graph-based smoothing on a landscape we can accurately evaluate without needing an oracle. Despite its simplicity, it is able to show how smoothing can lead to improved sampling of higher fitness sequences.
> > >
> > > > Additionally, I'm still unclear on how you derived the number 8 in the phrase 'up to 8 mutations' within the abstract.
> > >
> > > We derived 8 from our results on optimizing GFP (table 2) and AAV (appendix table 4) where our method is able to achieve improved fitness by applying on average 7-8 mutations from the starting training set in the hard difficulty. Our baselines do not come close to achieving similar extrapolation capabilities on this “hard” benchmark. We are happy to clarify or modify this statement if the reviewer still finds it unclear.
> > >
> > > > As an emergency reviewer not well-versed in this field, I didn't identify any significant technical flaws in this paper. Consequently, I recommend acceptance, though my confidence in this recommendation is low.
> > >
> > > We are grateful for the reviewers’ time to evaluate our work, especially for the keen questions despite it not being their expertise :).

---

### Author Rebuttal · Authors · 2023-08-10

We sincerely thank the reviewers for their constructive feedback. Our manuscript has improved as a result. We are glad reviewers noted the novelty of graph-based smoothing and Markov Chain Monte Carlo (MCMC) to protein fitness optimization. Here we highlight important improvements and responses to reviewer concerns

**Evaluation.** We understand concerns of using a trained neural network as the fitness score. However, many published works on protein engineering follow this practice and do not have experimental validation [1-5]. We believe the same standard should be applied to our work. Furthermore, we improved benchmarking over prior works in the following ways:

* **Improved oracle over the oracle [6] used in prior works**. We show in appendix figure 6 the improved robustness and spearman rho of our oracle.
* **Improved data splits based on difficulty.** We discovered data leakage in the training set of the previously used Green Fluorescent Protein (GFP) benchmark [6]. Our splits avoid data leakage, are significantly more challenging and provide greater insight in comparing methods.

**Method simplification.** We recently found that bi-level sampling is not necessary to achieve state-of-the-art in our experiments. Our updated method, called **Gibbs with Graph-based Smoothing (GGS)**, uses pointwise mutations yet is still able to discover multiple beneficial mutations. Specifically, instead of sampling from the distribution in equation (2), we sample both the location and substitution at once -- see equation (2) in the PDF. We have updated our manuscript accordingly to reflect this simplification; however, as shown in the table below, our analysis and conclusions have not changed.

| Task       | GFP_Easy   | GFP_Med   | GFP_Hard   | AAV_Easy   | AAV_Med   | AAV_Hard   |
| ---------- | ---------- | --------- | ---------- | ---------- | --------- | ---------- |
| Method     | GGS   BiGGS  | GGS BiGGS | GGS BiGGS  | GGS BiGGS  | GGS BiGGS | GGS BiGGS  |
| Fitness    | **0.93** 0.92 | **0.9** 0.87 | **0.81** 0.77 | **0.62** 0.61 | **0.62** 0.58 | 0.43 **0.46** |
| Diversity  | 2.0 2.0       | 2.7 3.2   | 2.6 2.6   | 2.7 3.0     | 2.5 4.5   | 8.3 9.8   |
| Novelty    | 1.0 1.0       | 5.4 5.8   | 7.0 6.5     | 1.0 1.0       | 5.0 4.0       | 8.0 5.0      |

**Intuition with Toy Example**
Protein fitness landscapes are difficult to analyze; therefore, we demonstrate the intuition behind graph-smoothing in a toy example using synthetic data where we can control the fitness landscape. Consider a simplified problem of optimizing a length-8 sequence $x \in V^8$ with vocabulary $V = \{A, B, C, D, E, F, G, H\}$ of size 8. Our rugged fitness landscape is defined with a noisy fitness function: $f(x) = tr(Ax)$ where $A=I + Z$ with a fixed gaussian noise matrix $Z$, identity matrix $I$, and $x$ represented as a one-hot matrix. The landscape is visualized in figure 1 of our attached PDF. The global optimum is the sequence $ABCDEFGH$ but many local optimums exist.

We construct a training set of 10000 random sequences at least 7 mutations away from $ABCDEFGH$ in order to sufficiently obscure the signal. The number of possible sequences is $8^8$ meaning the training set covers less than 0.1% of the full landscape. We decrease the hidden layer size of the CNN from 256 to 64, the number of eigenvectors in graph-based smoothing from 50 to 2, and increase the sampling temperature from 0.01 to 0.1. All changes reflect the large reduction in dimension from GFP/AAV and overall complexity to this toy example. All other hyperparameters remain the same.

Examining the table below, calculated as in the manuscript where fitness is min-max normalized with respect to the training set, we see that smoothing increases median fitness across 5 seeds, although we do note that, unlike for protein datasets, the unsmoothed model performs strongly as well. We attribute this performance to the lower complexity of this toy example.
| Predictor   | Unsmoothed | Smoothed |
| -------- | ---------- | -------- |
| Median Fitness  | 0.85       | **0.89**     |
| Diversity| 4.5        | 4.8      |
| Novelty  | 2.0        | 2.2      |

Figure 2 in the PDF displays a sample mutation trajectory generated by GGS using the smoothed predictor to show how smoothing can avoid local optimums. Denoting the sequences in the trajectory as $x_0, x_1, x_2, x_3, x_4, x_5$,  $x_5$ is the fittest sequence, and $x_3$ is a local optimum, as $f(x_2) < f(x_3) > f(x_4)$. Remarkably, the unsmoothed predictor follows the noisy contours of the landscape, whereas the  the smoothed predictor increases monotonically from $x_3$ to $x_5$, effectively traversing this fitness "valley." We hope this analysis convinces reviewers that smoothing leads to improvements even when it deviates from the true noisy landscape. This toy experiment has been added to the appendix.

We thank the reviewers for their engagement and look forward to their responses.

[1] Angermueller, Christof, et al. "Model-based reinforcement learning for biological sequence design." International conference on learning representations. 2019.

[2] Brookes, David, Hahnbeom Park, and Jennifer Listgarten. "Conditioning by adaptive sampling for robust design." International conference on machine learning. PMLR, 2019.

[3] Jain, Moksh, et al. "Biological sequence design with gflownets." International Conference on Machine Learning. PMLR, 2022.

[4] Ren, Zhizhou, et al. "Proximal exploration for model-guided protein sequence design." International Conference on Machine Learning. PMLR, 2022.

[5] Trabucco, Brandon, et al. "Conservative objective models for effective offline model-based optimization." International Conference on Machine Learning. PMLR, 2021.

[6] Trabucco, Brandon, et al. "Design-bench: Benchmarks for data-driven offline model-based optimization." International Conference on Machine Learning. PMLR, 2022.

---

### Decision · Program_Chairs · 2023-09-21

**Decision:**

Reject

**Comment:**

The consensus is that while the submission does showcase intriguing results, there are several major concerns that hamper its acceptance for NeurIPS:

* Lack of Practical Evidence: One recurring theme across the feedback is the need for a concrete, real-world application of the proposed method. Including a case study or more tangible evidence would significantly enhance the paper's credibility and showcase the practical applicability of the method. The paper's heavy reliance on a black-box neural network to compute the fitness score undermines the reliability and interpretability of the results.

* Hyperparameter Justification & Method Complexity: The chosen hyperparameters appear arbitrary, lacking any evident justification. Furthermore, the method's complexity, especially with steps like retraining after smoothing, is deemed as potentially limiting its adaptability to other applications.

* Evaluation Challenges: The reliance on machine learning models as oracles, even if trained on experimental data, presents a fundamental evaluation challenge. The possibility that these models might represent fitness landscapes in a manner different from actual protein landscapes is a crucial point of contention. While it's acknowledged that evaluating protein sequence design algorithms is inherently challenging, the paper would gain from exploring more fitness landscapes from existing literature and offering comparative insights.